# Insights and Perspectives on Plant-Based Beverages

**DOI:** 10.3390/plants12193345

**Published:** 2023-09-22

**Authors:** Aneta Popova, Dasha Mihaylova, Anna Lante

**Affiliations:** 1Department of Catering and Nutrition, Economics Faculty, University of Food Technologies, 4002 Plovdiv, Bulgaria; popova_aneta@yahoo.com; 2Department of Biotechnology, Technological Faculty, University of Food Technologies, 4002 Plovdiv, Bulgaria; 3Department of Agronomy, Food, Natural Resources, Animals, and Environment—DAFNAE, Agripolis, University of Padova, 35020 Legnaro, Italy; anna.lante@unipd.it

**Keywords:** plant nutrition, dietary habits, beneficial nutrition, antioxidants, sustainability

## Abstract

The emerging demand for everyday food substitutes is increasing on a daily basis. More and more individuals struggle with allergies and intolerances, which makes it mandatory to provide alternatives for simple products like dairy milk. Plant-based beverages (PBBs) are currently trending due to the multiple diets that promote their consumption with or without a justification. PBBs can derive from various types of plants, not exclusively nuts. Some of the most well-known sources are almonds, soy, rice, and hazelnuts, among others. In view of the need for sustainable approaches to resource utilization and food production, novel sources for PBBs are being sought, and those include fruit kernels. The plant kingdom offers a palette of resources with proven bioactivity, i.e., containing flavonoids, phenolic acids, vitamins, carotenoids, and phenolics, among others. Many of these beneficial substances are water soluble, which means they could be transferred to the plant beverage compositions. The current review aims at comparing the vast number of potential formulations based on their specific nutritional profiles and potential deficiencies, as well as their expected health-promoting properties, based on the raw material(s) used for production. Special attention will be given to the antinutrients, usually abundant in plant-based sources.

## 1. Introduction

Plant-based food options are a very fast-growing market sector, which reduces greenhouse gas emissions. Worldwide, the catering industry is witnessing important events (Met Gala, 2021) being organized with a 100% plant-based menu. According to a report [1], the global plant-based beverage (PBB) market is predicted to expand with a compound annual growth rate of 13.1% from 2023 to 2030. 

Currently, PBBs can be preferred for several reasons: allergies, intolerances, and other food restrictions; diet preferences; health issues (high low-density lipoprotein); environmental issues attracting the followers of vegetarianism, veganism, and flexitarianism [2] (lower carbon footprint); and animal welfare [3].As milk alternatives, PBBs, are emulsions from the plant(s) of choice and water, which is an aqueous extract of the raw materials but not a juice obtained by squeezing fruits and vegetables [4]. They can sometimes be referred to as imitation or vegetal milks because they resemble the consistency, density, and overall appearance of dairy milk. However, in many countries, the label “milk” for plant-based alternatives is not allowed [5]. 

Dairy milk is extremely important to the human diet because it can meet several nutritional needs of the individual, namely minerals, vitamins, fats, and proteins. Among the functional properties that are attributed to milk, the stimulation of insulin, incretin, glucose inhibitory polypeptide, and the insulin-like growth factor 1 play an important role [6,7]. In addition, milk is the raw material for producing cream, ice cream, yogurt, cheese, and butter [5]. As regards the concept of the negative “environmental footprint” associated with milk, the results reported by Tessari et al. [8] showed that parameters like land use and greenhouse gas emissions (GHGEs) of milk are comparable to and sometimes lower than the ones of other vegetal products, if the comparison is carried out on the basis of essential amino acid content, because of the higher quality of animal proteins. 

Researchers propose [9] that the least favorable to the environment are the almond beverages because of water consumption and the rice beverages due to energy consumption. The water footprint of almonds places them in the top 10 not only in physical weight (L/kg) but also in nutritional energy content (L/kcal) [9]. Nonetheless, PBBs generally have a small impact in terms of land use per liter production, ozone depletion potential, and global warming potential. With that said, certain PBBs can perform better in terms of GHGEs but cannot be processed and fail in cooking performances, which makes them unsuitable in certain ways [10]. Additionally, the nutritional value of PBBs can be questionable in terms of getting as close as possible to dairy milk [11]. According to Sethi et al. [2], there is a nutritional imbalance of PBBs in comparison to bovine milk but also a presence of plant bioactive compounds; the authors further note that technological issues are always an open question.

Therefore, the objective of this study was to perform a critical evaluation of the literature on the important characteristics of plant-based beverages, along with their advantages and disadvantages, and to define future research areas in view of the sustainable food production cycle. 

## 2. Plant-Based Beverages

PBBs present various types, flavors, and range widely in market options. Their manufacturing process is rather long, especially compared to the one of dairy milk. Some of the steps include soaking, grinding, blanching (enzyme inactivation), centrifuging, thermal treatment (microbiological safety), and optional fortification. Figure 1 presents a simplified view of the processing steps of PBBs from farm to table.

Plant-based drinks are colloid systems that contain not only liquid but also solid particles. This can threaten their stability and lead to sedimentation [12]. Food colloids continue to present a topic for research not only in terms of aggregation prevention but also process, cost efficiency, and interactions in the gastrointestinal tract [13]. Paul et al. [14] have studied the possibility of producing spray-dried PBB powders for prolonged storage. 

The term sustainable PBB has relevance here, taking into consideration the impact each source (legume, nut, kernel, and cereal, among others) has on the finished product, and the ecofriendly technologies employed in the production. For example, the production of the coconut milk-like beverage is reported as sustainable due to the reincorporation of the waste (husk and shell) as biomass for energy [15]. However, large industrial production is often associated with increased food miles. Food miles should be carefully incorporated into the sustainability evaluation. Their impact can be reduced only when local products are being used. In the case of PBB production, this cannot always be applied [16]. Recent literature has also focused on the possibility of plant-based usage in the production of fermented beverages [17]. Fermentation is an old technology that has proven its benefits over time [18]. Future research papers should continue to focus on its application to plant-based products.

### 2.1. Types and (Anti-)Nutritional Profiles

The current availability of PBBs gives a variety of options to choose from: drinks fortified with minerals, vitamins, or added protein, flavored or sweetened ones, and from different sources. An Austrian food tech startup used fruit-kernel products in line with the circular economy and sustainable usage of raw materials (fruit pits) [19]. This sets a trend for a sustainable approach for by-products and the production of PBBs, which are increasing in market presence. A bottle of cherry pits comprises 10,023 kernels; apricots—1333 kernels; plums—2356 kernels. A set of research should be performed in order to characterize these rather novel products, bearing in mind that fruit kernels are seen as waste products and can sometimes be implemented in cosmetology but are seldom used in food technology. Another sustainability evaluation can shed light on the resources used in the production of PBBs, with special attention given to the waste generated by PBB production. Future research might be able to provide information about the chemical composition of this waste, and its possible utilization. 

More common representatives of PBBs are legumes, nuts, cereals, and pseudocereals (i.e., quinoa, chia, amaranth), among others. Figure 2 summarizes several types of plant-based beverages in terms of range and source. In terms of flavor, the most frequent is vanilla, followed by chocolate and strawberry. Interestingly, the vanilla flavor is rarely natural in the food industry, and is most often synthetic/artificial [20]. This may be a controversial issue for vegans, for example, or for those in favor of natural foods.

The sweetening of PBBs is carried out with sugar, which can change the overall nutritional score of the beverage to a lower one, for example, from A to B, or from B to C. The Nutri score is used in the European Union to illustrate the healthiness of food products [21]. Each letter is also color-coded like a traffic light. The A and B scores are marked green, while the others range from yellow to red (C to G). The system evaluates ingredients that should be consumed cautiously, i.e., sugar, salt, and saturated fatty acids. 

The unsweetened variations are marked by naturally occurring sugars and no added sugar. All PBB assortments have the vegan sign on their packaging. Most of them are also marked lactose-free. This is in line with Regulation (EC) 1924/2006 on the nutrition and health claims in the European Union.

The nutritional value of PBBs has been subjected to research in several papers, some of which compare it to that of dairy milk, i.e., bovine. The topic has often been explored, especially in terms of infant and toddler nutrition [22] in view of the emerging exclusive plant-based diets. What should be taken into consideration is not only the macro- (protein, fat, carbohydrate) and micro- (minerals, vitamins) nutrient profile but also the availability of phytocomponents, which are highly beneficial to one’s health. 

According to research, rice beverages have the highest energy content of 56.8 ± 6.3 kcal/100 mL, depending on brand [22]. Other representatives like almonds, cashews, and coconuts, among others, can vary significantly in energy values depending on content and especially the addition of sugar as a sweetener [23].

The protein content can also vary widely [24]. For example, a sesame beverage can contain between 0.6 and 5.5 g/100 g protein, while sheep milk contains 3.35 g/100 g [3]. This proves that PBBs can meet the protein requirement when only numbers are considered. As previously reported, what is important is to evaluate the amino acid distribution in the PBB source and distinguish the limiting amino acids. Plant proteins are known for their lower quality compared to animal proteins [25]. However, PBBs are often regarded as sources of protein only when the latter is being added to the initial beverage. On the other hand, some PBBs, like the coconut ones, are marked as having 0 g/100 g of protein content [3]. When considering plant-based nutrition, special attention is paid to the protein quantity and quality in the daily diet. The availability of protein is lower in most plant-based foods and there are frequently several limiting amino acids. However, a suitable combination of plant sources can support a quality protein intake in terms of essential amino acids. For this reason, it is necessary to highlight certain plants as suitable substitutes not only in terms of rheological and textural properties but also nutritional provision. Plant proteins usually comprise globular proteins that are reported to act as good emulsifying and foaming agents [26]. For example, if albumins and globulins are being evaluated, peas and chickpeas have lower overall emulsifying properties compared to kidney beans, which possess good foamability and better emulsification properties [27,28]. Globular proteins can also be employed as gelling agents forming isotropic or anisotropic gels [26]. Flexible and fibrous proteins do not have a good plant analog compared to animal sources. For example, the absence of casein and collagen can present a challenge for the food industry [29,30]. From a nutritional perspective, soy, pea, grains, nuts, and legumes contribute more to the protein intake compared to pseudocereals [31].

The fatty acid (FA) composition of PBBs is generally comprised of long-chain FAs [32]. This leads to the contribution of polyunsaturated fatty acids (PUFAs), which are beneficial for cholesterol management, i.e., low-density lipoprotein. PUFAs are also considered important when evaluating the risk of coronary heart disease mortality [33]. They can control a number of homeostatic and inflammatory processes [34]. Some of the more-often-studied ώ3 fatty acids are eicosapentaenoic, docosahexaenoic, and α-linolenic acid. Plant-derived bioactive peptides are also examined for their antioxidant, antidiabetic, and anti-inflammatory properties, among others [35]. The FA availability and composition of PBBs will depend on the source, as nuts will contribute more compared to beans [36]. Nuts are reported to have a high FA content, which can influence the fat-soluble bioactive compounds’ availability [37]. The presence of myristic, palmitic, and stearic acids was documented in PBBs [38]. The same study revealed that the FA content may vary when the source has been germinated. The most significant changes were observed for the chickpea beverages. Only coconut drinks have short- and medium-chain fatty acids [39]. The trans fatty acid contents vary from 0.0015 g/100 g (coconut) to 0.022 g/100 g (hemp) drinks [39]. The fat content of PBBs can be evaluated as beneficial, especially in cases where a lower fat intake is advised.

The carbohydrate content of PBBs should be divided into two relative groups: sugars, and starch and fibers. The predominance will most often be of sugars, each time depending on the source of the beverage. For example, the carbohydrate contents of nut (almonds, cashews, hazelnuts), seed (flax, birdseed, hemp), and legume (soy) beverages are rather similar, as reported in the literature [4]. The sugars identified in PBBs were sucrose, fructose, glucose, and lactose [40]. Researchers stated that spelt, rice, and oat drinks had high total carbohydrate contents; coconut and cashew drinks—moderate; and soy, almond, and hemp drinks—low [39]. Fiber is rarely present in the nutritional content as it is very low. When it comes to starch, rice drinks are reported to have the highest amounts [39]. The carbohydrate content of PBBs can be seen as potentially provoking a higher glycemic response compared to bovine milk, as stated in a recent review [40]. The same paper outlined soy drinks as having a glycemic index most similar to dairy milk. The abovementioned works open up an interesting area of research given the manifestation of diabetes as a socially important non-communicable disease and how the need for proper glucose control is rising daily. Another noteworthy consideration is the ability of carbohydrates to alter the foaming and emulsifying properties of whey proteins. In view of plant-based nutrition, such topics should be studied more extensively. Furthermore, both proteins and polysaccharides can be responsible for better food stability, texture, and shelf life [41].

Considering the vitamin contents in PBBs, most published papers advocate the same idea. PBB are fortified with vitamins from the B group, vitamins A and E in combination with D and vitamin C [4]. For example, many nuts provide vitamin E, which is important to the human body as it cannot be otherwise synthesized [42]. However, in practical terms, the vitamin contents of PBBs can be variable, depending on the initial source. Furthermore, sometimes, water-soluble vitamins can be present in the PBB source, but their availability is significantly lowered during the initial soaking process. The synergy of antinutrients should also be taken into account [43]. When minerals are being evaluated, calcium is the focus, as bovine milk is a natural source of calcium. Thus, calcium fortification of PBBs is a very common practice. Calcium fortification can be achieved with carbonate, phosphate, or a mixture of both. Nevertheless, the calcium fortification is not usually comparable to the calcium available in bovine milk [44]. 

In order to illustrate the available labeled nutritional information, Table 1 summarizes the nutritional information of commercially available plant beverage assortments that are well-distributed worldwide. Nutritional labels are a trustworthy source of information, according to research [45]. They are seen as a tool for making an informed choice, although their success can vary between consumers. The nutritional label can provide simple, well-structured, and easy-to-compare information in order to make healthier buying decisions [46].

The products in Table 1 represent a world-known brand of PBBs that has focused on 100% plant-based drinks. It currently has a portfolio of plant-based drinks, barista options, alternatives to yogurt, and cream, among others. The PBB options range from original to no sugar, flavored, light, and early years (1–3+ years old). The calcium, reported in all drink variations, is introduced as tri-calcium phosphate (E341 food additive). The presence of nanoparticles in foods has become quite frequent in modern life [47]. The coconut beverage has the least energy value but contains the most saturated Fas. The vitamin and mineral addition is practically the same across beverages, while the presence of fiber is more noticeable in the soya, hazelnut, and oat drinks. All product variations are with naturally occurring sugars, without added sugar. 

Table 2 is a visual presentation of other PBB product options marketed in Lidl stores worldwide in line with their sustainability strategy to “make good food accessible to everyone”. 

A comparison between brands shows similarity in values, although the values stated on the packaging in Table 1 present more detailed nutritional information compared to the ones in Table 2. However, differences in packaging labels may occur in different geographical areas based on the laws and regulations that control the food market. In terms of protein, none of the assortments can account for a proper daily protein intake. The fat intake can be seen as more beneficial due to the presence of polyunsaturated Fas, and less saturated Fas. By default, carbohydrates are the easiest of the macronutrients in terms of sufficient intake; thus, the amount of carbohydrates can be seen as principally non-problematic. The point here is that carbohydrates are almost entirely presented by sugars, which are rapidly absorbed by the body. 

The antinutritional profile of PBBs should also be pointed out as the initial beverage sources are abundant in phytic acids, tannins, saponins, and enzyme inhibitors, as reviewed in the literature [48]. They can act by not only as having unpleasant effects on the gastrointestinal tract but also preventing the absorption of vitamins and minerals. However, some of the steps in the production process of PBBs (i.e., soaking, thermal treatment) can eliminate/lower the quantity of antinutrients and disable the exhibition of their harmful properties. Antinutrients are never presented in the label information and only a few people will know what they stand for and how to evaluate their importance and relevance. Thus, antinutrients should be better explained to the consumer.

Another important topic is the ability of PBBs to successfully substitute dairy milk for infants and toddlers. Research revealed that fortified (vitamins A and B12, calcium, zinc, and iodine) versions of soy drinks can be seen as acceptable variations [49]. This confirms that potential nutritional deficiencies can be overcome. However, a recent study [50] stressed that PBBs were particularly challenging for general practitioners because infants and toddlers demand proper nutrition for growth. In this view, the use of PBBs should be carefully addressed as deficiencies may arise. 

### 2.2. Health-Promoting Properties

The consumption of plant-based beverages has been shown to provide health benefits in terms of cardiovascular health, oxidative damage, and gastrointestinal health, among others [51,52]. 

Polyphenols, along with phytosterols and carotenoids, are important phytochemicals, part of the human diet [53]. Their bioavailability can be relatively low but they still interact in the body and provide biological effects [54]. Some of the most studied polyphenols are phenolic acids, flavonoids, and anthocyanins [55]. Flavonoids are structured as flavanones, flavones, isoflavones, and flavonols [56]. Flavonoids, along with their biological activity, help with the treatment of type 2 diabetes, can protect against certain types of cancer, and exhibit anti-inflammatory immunomodulatory and hepatoprotective effects [57]. Some of the most-often-listed phenolic acids in the literature are caffeic, vanillic, p-coumaric, ferulic, carnosic, rosmarinic, and gallic acids [58]. Plant sterols are particularly effective in lowering LDL cholesterol [59]. Carotenoids are related to the management of ocular diseases, as well as some types of cancer and cardiovascular ailments [60]. 

Many researchers have focused on identifying various biologically active molecules in plant-based matrices. Particular information about the content of phytochemicals in PBBs is missing, even though such information has been provided for the sources of the PBBs. 

For example, oats are reported to contain several phenolic acids (ferulic, vanillic, caffeic, protocatechuic, and sinapic), sterols, and avenanthramides [61]. Oat saponins aid in cholesterol lowering and immunoregulation [62]. However, the content of phytic acid should be carefully evaluated as it can play an antinutritional role by lowering the absorption of some vitamins and minerals [63]. Phytic acid is also found in sesame seeds, which are otherwise nutritionally beneficial. They contain polyphenols, phytosterols, aldehydes, anthraquinones, naphthoquinones, and triterpenoids, among others [64]. Soybeans can be seen as significant sources of protein for plant eaters, and the soy beverage is the one with the most protein (Table 1 and Table 2). Additionally, soybeans contain phenolic acids, flavonoids, isoflavones, saponins, phytosterols, and sphingolipids [65]. The same authors state that due to the presence of those phytochemicals, antioxidant, antidiabetic, antihyperlipidemic, and anti-obesity activities could be exhibited. Sphingolipids play an important role in immunity and inflammation management [66]. Tree nuts (almonds and pistachios) are rich in phytochemicals, i.e., polyphenols, carotenoids, and phytosterols [67]. These bioactive molecules can aid in the exhibition of anticancer, antioxidant, and antimicrobial activities [68]. Macadamia nuts are reported to have a change in the composition and availability of some of their compounds during roasting, especially those with positive health attributes (lipophilic compounds and phenolic compounds) [69]. Pistachio nuts can supply xanthophyll carotenoids and a wide spectrum of bioactive phenolic compounds [70]. The Brazil nut is reported as a food matrix rich in nutrients [71]. Nuts, seeds, and pulses are by default nutrient-dense foods with quality plant protein, and rich mineral and vitamin content [72]. Pulses have been established to possess extensive functional properties for food applications and are used to substitute animal proteins [73]. Since polyphenols are the most widespread bioactive compounds in agroindustry by-products of fruits, seeds, cereals, nuts, and vegetables, it is essential to recover these compounds to produce functional foods and Ingredients [74]. 

All of the above sources can be used to produce PBBs. This sets the scene for researchers to seek potential health-promoting properties provided by phytochemicals. It would be of interest to design studies that focus on the evaluation of phytochemicals in PBBs and their possible beneficial effects on human health since many biological activities are reported in the literature. In this respect, ecofriendly technologies such as ultrasound and the use of enzymes and fermentation could improve the traditional plant-based beverage processes, increasing the recovery efficiency of plant bioactive compounds, as well as the nutritional, sensory, and functional characteristics of the PBBs [75]. 

## 3. Materials and Methods

This review was conducted using the Scopus and PubMed databases. The search for articles was structured in a timeframe of 10 years (2013–2023) and we utilized a set of keywords (plant-based beverages, sustainable). The nutritional value was checked by label availability on market options of two widespread companies. For the nutritional profile documented in articles, the following have been included: nutritional value, minerals, vitamins, phytocomponents, and fortification. All published papers should have availability in English and free access to content. These prerequisites narrowed the articles to 89 in Scopus and 175 in PubMed (counting duplicates in databases published by July 2023) (Figure 3). The information was then critically structured for this review, referencing 78 sources. 

## 4. Conclusions and Future Perspectives

PBBs have gained popularity in recent years not only due to a need for more sustainable everyday living but also as they provide options for individuals with health conditions, i.e., allergies and intolerances. A growing trend for plant nutrition has seen various types of PBBs introdued. Their nutritional values have been reported in the literature, in comparison often to bovine milk. Results have shown that the availability of nutrients can be sufficient for a healthy diet in some of the available variations like soy drinks, while attention has to be paid to some other drinks like the coconut one. 

The presence of bioactive molecules in PBBs can support its frequent use, although antinutrients should not be neglected. PBBs will continue to gain market share as companies seek new sources for production. A future research challenge is presented in finding ways to fully incorporate PBB in recipes with thermal processing. The existing nutritional profile of PBBs has to be enriched, with the current missing information mostly on phytochemicals, vitamins, and minerals. Additionally, the full environmental impact has to be thoroughly evaluated.

## Figures and Tables

**Figure 1 plants-12-03345-f001:**
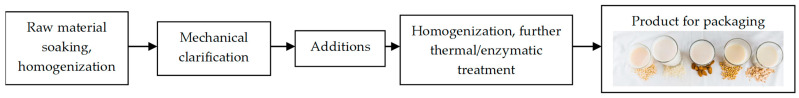
Processing steps of PBB manufacturing.

**Figure 2 plants-12-03345-f002:**
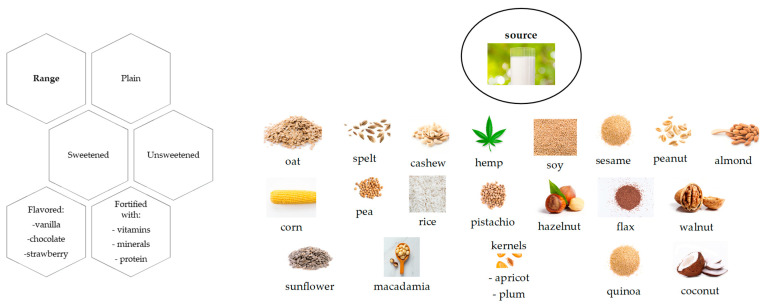
Types of PBBs.

**Figure 3 plants-12-03345-f003:**
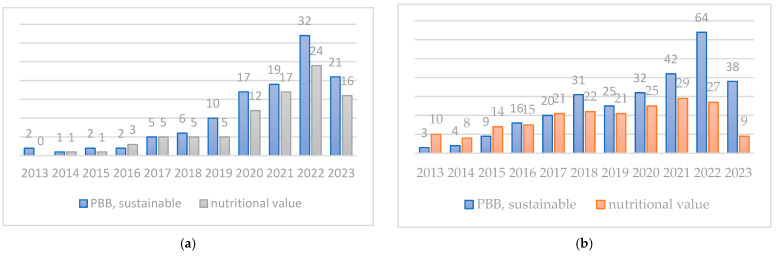
Database results of searches in (**a**) Scopus and (**b**) PubMed.

**Table 1 plants-12-03345-t001:** Nutritional values of commercially available plant beverage assortments per 100 mL.

Type of Product	Protein, g	Carbohydrates *, g	Sugars *, g	Fiber *, g	Fats *, g	Saturated Fas *, g	Energy Value *, Kcal	Vitamins *, Minerals *
Oat Original	0.3	7.2	3.3	1.5	1.5	0.1	46	D (0.75 µg); B2 (0.21 mg); B12 (0.38 µg); calcium (120 mg)
Hazelnut Original	0.4	3.2	3.2	0.3	1.6	0.2	29	A (0.12 mg); D (0.75 µg); E (1.8 mg); B2 (0.21 mg); B12 (0.38 µg); calcium (125 mg)
Rice Original	0.1	9.5	3.3	0	1	0.1	47	D (0.75 µg); B12 (0.38 µg); calcium (120 mg)
Cashew	0.5	2.6	2	0.2	1.1	0.2	23	D (0.75 µg); E (1.8 mg); B2 (0.21 mg); B12 (0.38 µg); calcium (120 mg)
Soya Original	3.3	2.7	2.5	0.6	1.9	0.3	42	D (0.75 µg); B2 (0.21 mg); B12 (0.38 µg); calcium (120 mg)
Coconut Original	0.1	2.7	1.9	0.1	0.9	0.9	20	D (0.75 µg); B12 (0.38 µg); calcium (120 mg)
Coconut and Almond	0.3	2.5	2.5	0	1.3	0.6	24	D (0.75 µg); E (1.8 mg); B2 (0.21 mg); B12 (0.38 µg); calcium (120 mg)
Almond Original	0.4	2.4	2.4	0.4	1.1	0.1	22	A (0.4 mg); D (0.75 µg); E (1.8 mg); B2 (0.21 mg); B12 (0.38 µg); calcium (120 mg)

* Values correspond to the information stated on the packaging.

**Table 2 plants-12-03345-t002:** Nutritional value of plant beverage assortments per 100 mL.

Type of Product	Protein *, g	Carbohydrates *, g	Sugars *, g	Fiber *, g	Fats *, g	Saturated Fas *, g	Energy Value *, Kcal
Oat	0.4	5.6	4.4	N/A	1.2	0.1	37
Hazelnut	0.6	3.4	3.2	0	2.1	0.2	36
Almond	0.5	2.8	2.7	0	1.2	0.1	24
Oat Barista	1.1	6.7	1.6	N/A	3.4	0.3	63
Coconut and Rice	0.1	2.7	1.9	0.1	1.3	1.2	23
Soya	3.3	2.9	2.9	0.8	1.9	0.3	44
Spelt	0.9	8.4	6	0.5	0.7	0.1	45
Rice	0.1	9	2.6	0	1	0.1	45

* Values correspond to the information stated on the packaging; N/A—information not available.

## Data Availability

No new data were created or analyzed in this study. Data sharing is not applicable to this article.

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
