# Peer review of "Insights and Perspectives on Plant-Based Beverages"

_plants, 2023, doi:10.3390/plants12193345_

Round 1

Reviewer 1 Report

This article is very interesting and will certainly add to the growing literature base on plant-based beverages. This is a review article; while the authors provide plenty of information there are a few suggestions and comments that may help them enhance the readability of the paper.

1. p 3 line 85- explain "plenty of opportunities".

2. Between lines 84 and 92- the authors appear to be attempting to describe the source of raw ingredients for the production of PBB. However, the listing of the number of kernels in a bottle does not lead to this explanation. Instead it may be better to clarify which part of the plant is used for the production of the beverage, what is the wastage from the raw materials and what happens to the waste-this may be important from a sustainability perspective.

3. Line 125- "initial content of what"

4. Line 93- explain pseudocereals.

5. Lines 137-139- Flexible and fibrous proteins do not have a good plant analogue compared to animal sources which presents a challenge 138 for the food industry [25,26]- Which flexible and fibrous proteins are being referred to? should source replace analogues? I would think plant proteins are better sources of fibrous proteins- not clear at all.

The above is an example of clarification that is needed throughout the paper-authors need to pay attention to language and what is being conveyed to the reader.

6. Line 148- What is the contribution of nuts? details need to be provided.

7. Line 230-231 edits language- The consumption of plant-based beverages has been shown to provide health benefits in terms of cardiovascular health, oxidative damage, gastrointestinal health, among others.

Again, this is an example of language that needs to be clarified for ease of reading and appreciating and understanding the benefits of PBB.

8. The authors state-the objective of this paper is to perform a critical evaluation of literature on the important characteristics of plant-based beverages with their advantages and disadvantages, and to define future research areas in view of the sustainable food production - some questions to consider (a) how is sustainability being defined? there is not much information on sustainability of PBB production-this needs to be written clearly within the contextual definition of sustainability- for example what would the sustainability be of coconut milk production and usage in the western world where coconuts are not produced. 

9. Explain more clearly the issues surrounding the use of PBB for infant nutrition-pros and cons. This is important especially from a consumer use point of view and how nutrition professionals should present this information.

10. Obviously PBB have the distinct advantage of contributing phytochemicals that are not evident in bovine milk- while it is important to delineate which phytochemicals exist in various PBB, it is equally important to consider how the processing can impact the availability of phytochemicals. There is use of thermal processing, how would this impact certain phytochemicals for example.

Please edit the entire paper for English language use and grammatical edits.

Reviewer 2 Report

The ever-evolving landscape of dietary preferences and the increasing prevalence of allergies and intolerances have led to a remarkable shift towards plant-based alternatives, especially in the realm of everyday food items. In this context, the article "Insights and Perspectives of Plant-Based Beverages" sheds valuable light on the burgeoning trend of plant-based beverages (PBB) and their profound implications for our health and the environment.

The article opens by acknowledging the growing demand for dairy milk substitutes, a demand that is not merely driven by dietary restrictions but is increasingly fueled by conscious choices to promote sustainability and well-being. Plant-based beverages have, without a doubt, assumed a pivotal role in addressing this demand, and this review undertakes the commendable task of providing a comprehensive exploration of this fascinating subject.

One of the article's strengths lies in its meticulous examination of the diverse sources from which PBB can be derived. Notably, it emphasizes that PBB are not confined solely to nut-based options; rather, they extend to an array of choices, including almond, soy, rice, hazelnut, and even fruit kernels. This inclusivity highlights the versatility of plant-based sources, making them a promising avenue for catering to diverse dietary preferences.

The article goes further by delving into the nutritional aspects of PBB. It highlights the presence of a range of bioactive compounds, such as flavonoids, phenolic acids, vitamins, carotenoids, and phenolics, many of which are water-soluble. This discovery not only underscores the potential health benefits of PBB but also hints at their capacity to be enriched with these beneficial substances. This aspect of the article offers valuable insights into the nutritional profile of PBB and their potential to positively impact human health.

A notable aspect of the review is its balanced consideration of antinutrients, which are often abundant in plant-based sources. This acknowledgment of potential drawbacks demonstrates the article's commitment to providing a well-rounded assessment of PBB, ensuring that readers are informed about both the advantages and challenges associated with their consumption.

In conclusion, the article "Insights and Perspectives of Plant-Based Beverages" admirably consolidates a wealth of information on this contemporary and significant topic. It underscores the rising popularity of PBB, not only as a sustainable dietary choice but also as a viable option for individuals with specific health requirements. The comparative analysis of different PBB types against bovine milk is particularly informative, revealing variations in nutrient availability that can guide consumers in making informed choices.

The article also raises important questions about the future of PBB. It anticipates a continued increase in their market share as companies explore new sources for production, and it highlights the need for further research to fully integrate PBB into recipes with thermal processing. Additionally, the call for enriching the nutritional profile of PBB with detailed information on phytochemicals, vitamins, minerals, and a thorough evaluation of their environmental impact emphasizes the article's forward-thinking approach.

In summary, "Insights and Perspectives of Plant-Based Beverages" is a commendable contribution to the field of nutrition and sustainability. Its well-researched content and forward-looking perspectives make it an essential read for anyone interested in the future of food and beverage consumption. This article not only informs but also inspires further exploration of the promising world of plant-based beverages.

Author Response

The authors would like to show their appreciation for the reviewer’s helpful and exceptionally positive review.

We very much hope that the manuscript will be accepted for publication.

Reviewer 3 Report

The paper is interesting and comprehensive offering insights regarding the comparison of nutritional composition of the plant-based beverages found on the market. The recent literature in the field was also consulted and referred in the manuscript. Still, I belive it will be useful for the reader also a comparison of the process technology (fermented/not fermented PBB, emerging technologies that might help increase the nutritional composition, etc.). 

In recommend the paper publishing after its minor revision.

Some small spelling errors were encountered. Please check the manuscript carefully. 

Round 2

Reviewer 1 Report

My comments surrounded around the use of the English language in certain sentences for clarity; more explanation on what the pros and cons are for the use of PBB for infant feeding- there is paucity of info ration but existing guidelines that the authors do not contextualize in this paper. This is important especially since PBB cannot replace bovine milks completely unless other aspects of the diet are considered too. Bodnar LM, Jimenez EY, Baker SS. Plant-Based Beverages in the Diets of Infants and Young Children. JAMA Pediatr. 2021;175(6):555–556. doi:10.1001/jamapediatrics.2020.5840

Need improvement
